# Structural Characterization and Digestibility of Curcumin Loaded Octenyl Succinic Nanoparticles

**DOI:** 10.3390/nano9081073

**Published:** 2019-07-26

**Authors:** Zhongshan Hu, Tao Feng, Xiaolan Zeng, Srinivas Janaswamy, Hui Wang, Osvaldo Campanella

**Affiliations:** 1School of Perfume and Aroma Technology, Shanghai Institute of Technology, No. 100 Hai Quan Road, Shanghai 201418, China; 2Department of Dairy and Food Science, South Dakota State University, Brookings, SD 57007, USA; 3Biological Engineering and Whistler Carbohydrate Research Center, Philip E. Nelson Hall of Food Science, Room 2195745 Agricultural Mall, Purdue University, West Lafayette, IN 47906, USA; 4Department of Food Science and Technology, The Ohio State University, 2015 Fyffe Road, Columbus, OH 43210-1007, USA

**Keywords:** curcumin, short glucan chains, octenyl succinic anhydride, amphiphilic biopolymer, in vitro digestion

## Abstract

Curcumin displays anti-cancer, anti-inflammatory and anti-obesity properties but its water insolubility limits the wholesome utility. In this study, curcumin has been encapsulated in an amphiphilic biopolymer to enhance its water solubility. This was accomplished through self-assembly of octenyl succinic anhydride–short glucan chains (OSA–SGC) and curcumin. The nanoparticles were prepared with the degree of substitution (DS) of 0.112, 0.286 and 0.342 of OSA. Thus prepared nanoparticles were in the range of 150–200 nm and display high encapsulation efficiency and high loading capacity of curcumin. The Fourier-transform infrared (FTIR) and X-ray diffraction analyses confirmed the curcumin loading in the OSA–SGC nanoparticles. The complexes possessed a V-type starch structure. The thermo gravimetric analysis (TGA) revealed the thermal stability of encapsulated curcumin. The OSA–SGC nanoparticles greatly improved the curcumin release and dissolution, and in-turn promoted the sustained release.

## 1. Introduction

Starch is a key energy source for humans and is an easily acquired and highly nutritional biopolymer. It displays excellent biodegradability and non-toxicity [1]. It has been widely used in a variety of food products and industrial applications [2]. It is composed of amylopectin and amylose, and variations in the amounts result in a variety of starches such as waxy corn, rice, wheat, etc., to name a few. Waxy cornstarch is predominantly composed of amylopectin (95–99%) [3,4]. Although waxy cornstarch had been identified as a useful biomaterial, its water insolubility restricts the widespread application. In this regard, addition of groups such as octenyl succinic anhydride (OSA) on the starch chains is being explored [5]. OSA starches displays excellent emulsification properties and good encapsulation efficiency toward many sensitive and insoluble functional molecules [6]. Amylose is also employed as a functional material, examples include bovine hemoglobin encapsulation in amylose matrix as an oxygen carrier, but not as flexible as short glucan chains (SGC). Starch could be modified by pullulanase to produce short glucan chains. Pullulanase is a well-known starch debranching enzyme [7], and around 70–80% of starches are being treated with pullulanase to generate short glucan chains mainly for encapsulation applications. Reddy et al. used pullulanase to modify corn, potato, cassava and other plant starch to form inclusion complex with stearic acid, and they found that enzymatic debranching of starch effectively improved the complexation, crystallinity, dispersion and stability of starch–stearic acid complexes. [8].

Curcumin is a plant phenol and makes up to 5% of the dietary spice turmeric possessing antioxidant [9,10], anti-inflammatory [11,12], anti-tumor [13], anti-HIV [14], anti-bacterial and anti-microbial [15] properties. In addition, curcumin could cure respiratory diseases and improve the human health. However, curcumin is water insoluble and is not readily absorbed in the human digestive system, in neutral, acidic and alkaline conditions. In addition, its photosensitive nature constrains the general utility and limits the product development. In this regard, encapsulating curcumin into amphiphilic polymers could circumvent the issue [16,17,18,19]. Yu [20] encapsulated curcumin into hydrophobically modified starch to enhance the in vitro anti-cancer activity. The OSA starch is also used to encapsulate water insoluble essential oil [21].

Herein, we prepared OSA–SGC–curcumin nanoparticles to improve the water solubility and bioavailability of curcumin. Thus prepared nanoparticles have been characterized by FTIR, XRD, TGA, TEM and dynamic light scattering (DLS). The effect of pH on the curcumin release kinetics has been established in the simulated intestinal and blood environments.

## 2. Materials and Methods

### 2.1. Materials

Waxy cornstarch was a gift from Ingredion Co., Ltd. (Guangzhou, China). Pullulanase (1000 ASPU/g) was obtained from Novozymes Investment Co. Ltd. (Bagsvaerd, Denmark). OSA (≥95.0%) was purchased from Tokyo Chemical Industry Co., LTD. (TCI, Tokyo, Japan). Curcumin (≥94.0%) was obtained from the Sigma-Aldrich Chemical Co. (St. Louis, MO, USA), and all chemical reagents were analytical grades.

### 2.2. Preparation of OSA Grafted on SGC and OSA–SGC Nanoparticles

The OSA–SGC was produced by the method described by Sun et al. [22] with modifications. Briefly, waxy cornstarch (15 g/mL) slurry (100 mL phosphate buffer solution, pH = 5.0) was kept in boiling water under vigorous stirring for about 30–40 min. The pullulanase (0.015 g/mL) was added and the sample was incubated at 8 ≤ pH ≤ 9, 58 °C for 6 h, and centrifuged at 1300× *g* for 2 min, and lyophilized for 48 h to produce SGC powder. Then 5 g of SGC powder was dissolved in 100 mL distilled water to form 5% (w/v) aqueous solution. SGC solutions were incubated in an oil bath at 121 °C for 30 min and 0.1 mol/L NaOH was added to keep the solution pH at around 8.5 to provide a suitable environment for OSA to make reaction with the SGCs. The OSAs (equivalent to 25%, 50% and 100% of the weight of SGC powder) were dispersed in about 2 mL anhydrous ethanol and added to the SGC mixture with continuous magnetic stirring with 8.5 ≤ pH ≤ 9.0 at 55 °C for 8 h, and then the pH was adjusted to 6.8 with 0.1 mol/L HCl solution to stop the reaction. The OSA–SGC were washed twice with 99.7% anhydrous ethanol and precipitates were cooled at 4 °C for 8 h before freeze-drying. The OSA–SGC polymers equivalent to 25%, 50% and 100% of the dry weight of SGC powder were recorded as OSA_0.25_–SGC, OSA_0.5_–SGC and OSA_1.0_–SGC, respectively. Finally, 100 mg of the above three OSA–SGC polymers were weighed and dispersed in a phosphate buffer solution of pH = 7.4 to prepare a 10 mg/L solution. After the solution was stirred in a constant temperature water bath at 37 ° C for 6 h, it was cooled to room temperature to obtain an OSA–SGC nanoparticle solution. The OSA–SGC nanoparticles were then precipitated with absolute ethanol and washed with water for 2–3 times. Finally, the precipitate was freeze-dried to obtain a dry powder of OSA–SGC nanoparticles.

### 2.3. Determination of Degree of Substitution

NMR was a common method to determine the degree of octenyl succinate substitution of the short glucan chain. This experiment was carried out by hydrogen spectroscopy (Brook, Steffisburg, Switzerland) at 600 MHZ, weighing 20 mg of octenyl succinate and short glucan chain, respectively. Of octenyl succinate short–glucan chain nanoparticles 20 mg were put in a nuclear magnetic tube, with the addition of 0.7 mL deuterated-dimethyl sulfoxide (DMSO-d6) containing 0.5% (w/w) LiBr. Then the mixture was heated at 80 °C to dissolve those nanoparticles, finally 20 mg deuterated-trifluoroacetic acid (TFA-d1) was added in the solution. A small amount of TFA-d1 could separate the peak of the hydroxyl group of the short glucan chain and the hydroxyl group in the water molecule, so that the nuclear magnetic spectrum was clearer. The formula for calculating the degree of substitution was as follows:(1)$$DS=\frac{I_{0.89}}{3(I_{\alpha -1,\;6}+\;I_{\alpha -1,\;4}+\;I_{r-e})},$$ where *I*_0.89_ represented the integral of the CH_3_ signal peak in OSA, *I*_*α*-1, 4_ represented the integral of the signal peak at 5.11 ppm of the proton on the α-(1→4) connected carbon atom, *I*_*α*-1,6_ represented the integral of the proton signal peak on the α-(1→6) connected carbon atom, *I_r-e_* corresponds to the reducing chain ends. *I_r-e_* was the ^1^H NMR integrals of the internal α and β reducing chain ends at approximately 4.28 and 4.91 ppm in a short glucan chain [23].

The reaction efficiency formula is as follows:(2)$$\mathrm{RE}\;=\frac{\mathrm{Actual}\;\mathrm{DS}}{\mathrm{Theoretical}\;\mathrm{DS}}\times 100\%.$$

Among them, the theoretical degree of substitution (DS) was based on the assumption that all added anhydride reacts with starch to form ester derivatives.

### 2.4. Determination of the Critical Micelle Concentration (CMC)

The critical micelle concentration (CMC) of the short glucan chain octenyl succinate solution was determined by the fluorescence probe method [24]. Firstly, 1 mg/mL OSA–SGC nano-micelle blank solution was prepared, and the original solution was diluted with distilled water into 0.5, 0.1, 0.025, 1 × 10^−3^ and 0.05 × 10^−3^ mg/mL. Pyrene/acetone solution with the concentration of 6 × 10^−5^ mg/mL was prepared. Of the pyrene/acetone solution 1 mL was taken in a beaker and stored in tin foil. The pyrene/acetone solution was heated at 40 °C for 30 min. Through adding 10 mL of the above different concentrations of nanomicelle solution into the pyrene/acetone solution, and the pyrene concentration in the solution was 6 × 10^−6^ mg/mL. By stirring at room temperature for 24 h, the pyrene/acetone solution was fully mixed. The fluorescence intensity of the solution was measured in a fluorescence spectrophotometer (F-2000, Hitachi, Tokyo, Japan) with an excitation wavelength of 335 nm, a scanning wavelength of 350 to 550 nm, and a slit width of excitation and emission just about 5.0 nm. The fluorescence emission intensity of pyrene at 383 nm in nanoparticles was recorded as I_3_, and the fluorescence emission intensity at 373 nm in water was recorded as I_1_. With the increasing concentration of pyrene in the solution of nanoparticles, the fluorescence at 383 nm will increase, while the fluorescence at 373 nm would be relatively weakened. With continuous changes of the concentration, the I_3_/I_1_ value of pyrene in water tended to be a constant, and the solubility of pyrene in nanoparticles would increase, and a mutation value would appear in I_3_/I_1_, indicating that the nanoparticles were formed at this time, and the concentration of the mutation point was the critical micelle concentration (CMC).

### 2.5. Preparation of Curcumin OSA–SGC Complexes

Curcumin was dissolved in ethanol at the concentration of 5 mg/mL, and OSA–SGC solutions were prepared with 10 mg/mL in PBS with 2 h heating according to Sun’s [25] method with modification, and were sonicated in an ultrasonic cleaner for 15 min. Curcumin solutions were slowly added to OSA–SGC with a ratio of curcumin: OSA-SGC 1:5 (v/v) and heated at 37 °C under constant stirring for 8 h. After that, rotary evaporation was used to remove ethanol at 38 °C for 30 min. The OSA–SGC–CUR (curcumin) were freeze-dried, and kept in a dryer for further analysis.

### 2.6. Encapsulation Efficiency (EE) and Loading Content (LC)

Column: Agilent HC-C18 (5 μm × 4.6 mm × 250 mm); temperature: 40 °C; injection volume: 10 μL; detection at 430 nm; flow rate; 1.0 mL/min. Mobile phase A: 5% glacial acetic acid water, mobile phase B: acetonitrile.

Curcumin was dissolved in 400 mL methanol at a concentration of 1000 μ/mL under constant stirring. The samples were divided into five parts with diluted concentrations of 1, 5, 10, 20 and 50 μg/mL. High-performance liquid chromatographic spectra were recorded with the set peak area as the abscissa and concentration as the ordinate: The curcumin standard curve equation was thus established. The OSA–SGC–CUR nanoparticle samples were added to methanol at a concentration of 1 mg/mL and shaken for 2 min. The peak area of the supernatant at t 430 nm was determined, as was the concentration of curcumin corresponding to the standard curve. OSA–SGC–CUR nanoparticle samples were added to methanol at a concentration of 1 mg/mL and mixed to release the curcumin under high-speed centrifugation (1500× *g*, 20 min). Then, the supernatants were reserved to measure the curcumin peak area at 430 nm, and the curcumin concentration calculated according to a standard curve.

(3)$$\mathrm{EE}(\%)=\;\frac{\mathrm{Total}\;\mathrm{amount}\;\mathrm{of}\;\mathrm{curcumin}-\mathrm{Total}\;\mathrm{free}\;\mathrm{curcumin}}{\mathrm{Total}\;\mathrm{amount}\;\mathrm{of}\;\mathrm{curcumin}}\times 100.$$

(4)$$\mathrm{LC}\%\;=\;\frac{\mathrm{Embedded}\;\mathrm{curcumin}\;\mathrm{content}}{\mathrm{The}\;\mathrm{weight}\;\mathrm{of}\;\mathrm{nano}\;\mathrm{micelles}}\times 100.$$

### 2.7. FTIR of OSA, OSA–SGC and OSA–SGC–CUR

The molecular structures of SGC, OSA–SGC and OSA–SGC–CUR were confirmed using FTIR spectrophotometry (c). Samples were spread on the countertop and scanned under ambient conditions, and the FTIR spectra of SGC, OSA–SGC and OSA–SGC–CUR were accumulated, from 400–4000 cm^−1^, on an FTIR spectrophotometer with 32 scans at a resolution of 4 cm^−1^. Data were processed by the Origin Pro 2017C Software.

### 2.8. X-ray Diffraction

The X-ray patterns of SGC, OSA–SGC and OSA–SGC–CUR were carried out using a X-ray diffractometer (D8 Advance, Bruker, Rheinstetten, Germany) with Cu Kα (1.5418 Å) radiation at a voltage of 40 kV and a current of 30 mA. One gram of the sample was dispersed in the sample holder. The scanning region of diffraction angle (2θ) was set from 5 to 50° with a scanning speed of 4°/min, and a time step of 0.4 s.

### 2.9. Dynamic Light Scattering (DLS)

The average size and zeta potential of the samples were measured by DLS, using a Malvern Nano Zetasizer (Malvern Instruments Ltd., Malvern, UK). The method was performed on samples diluted in distilled water and ultrasonicated for 5 min before measurement at a diffraction angles at 90° before analysis at 25°.

### 2.10. Thermogravimetric Analysis (TGA)

The thermal stability of OSA, OSA–SGC, and OSA–SGC–CUR was assessed by TGA. Thermogravimetric analysis was conducted with a Q-5000 (TA Instruments Inc., New Castle, DE, USA) instrument, was first calibrated with indium: The 2 to 5 mg inclusion compound samples were placed in a crucible and the temperature increased from 20 to 400 °C, at 10 °C/min with nitrogen used as a protective gas. The resulting data were analyzed using the Universal Analysis V4.5A Instrument software.

### 2.11. Transmission Electron Microscopy (TEM)

The transmission electron microscopy was performed using a Tecnai G2 F30-TWIN from Bruker, Germany, with an acceleration voltage of 200 kV. The OSA–SGC nanoparticle dry powder was formulated into a solution with a mass fraction of 1%, and sonicated at room temperature for 10 min, then dropped onto a copper mesh with a carbon support film, and was freeze-dried for measurement. The samples were kept in a sterile state and remained transparent.

### 2.12. Intestinal and Blood Environment Slow Release Simulation

Around 1 mL of OSA–SGC–CUR nanoparticle solution was packed in a dialysis bags and placed in a 30 mL phosphate buffer solution at pH 6.8 and 7.4 with 3% (v/v) Tween 80, pancreatin (57.6 mg/mL) and bile salts (46.8 mg/mL). The absorbance of curcumin was measured by the 1 mL external solution of dialysis bag, which was obtained by constant temperature oscillated at 37 °C at 200 rpm for 0, 2, 4, 6, 8, 10, 12, 14, 16, 18, 20, 25, 30, 35, 40, 45, 50, 55, 60, 65, 70 and 75 h. After measurement, pH 6.8 and 7.4 phosphate fresh buffer solutions, which contained 3% (v/v) Tween 80, pancreatin (57.6 mg/mL) and bile salts (46.8 mg/mL) should be supplemented in time to keep the total volume of 30 mL unchanged. The extracted solution was measured at 430 nm by a UV-vis spectrophotometer, and amount of dialyzed curcumin was calculated from the obtained concentration.

(5)$$\mathrm{Release}\;\mathrm{rate}\;(\%)\;=\;\frac{\mathrm{Curcumin}\;\mathrm{released}\;\mathrm{in}\;\mathrm{dialysate}}{\mathrm{Encapsulated}\;\mathrm{curcumin}}\times 100\%.$$

### 2.13. Statistical Analysis

All measurements were carried out in triplicate and average results are reported. Data were analyzed by an analysis of variance (ANOVA), followed by Duncan’s multiple range test using SPSS 22 Statistical Software Program (SPSS Incorporated, Chicago, IL, USA). A value of *p* < 0.05 was considered statistically significant.

## 3. Results and Discussion

### 3.1. Determination of the Degree of Substitution

In the Figure 1, the proton chemical shifts on α-1,4-glycosidic bonds ranged from 5.11 to 5.25 ppm. Compared with SGC, the proton vibration on glucose residues changed slightly. With the increase of the degree of substitution, the absorption peak at 5.11–5.25 ppm became wider, which was due to the fact that the reaction of octenyl succinic anhydride was mainly carried out at O-2, which had been reported by Bai [26] et al. However, the chemical shifts of protons on α-1 and 6-glycosidic bonds did not change significantly, indicating that waxy cornstarch was completely hydrolyzed by pullulanase to obtain SGC. The proton chemical shifts at the end of the reducing chain of SGC appeared about 4.30 ppm and 4.92 ppm. Compared with the SGC NMR spectrum (Figure 1B), the nuclear magnetic resonance (Figure 1C–E) of the esterified octenyl succinate short glucan chain appeared a new absorption peak in 0.89 ppm. This new absorption peak was caused by the influence of methyl at the acyl end, indicating that the esterified group had been successfully introduced into the short glucan chain. No residual signals of unreacted OSA were observed in the ^1^H NMR spectra around 3 ppm in Figure 1C–E. The chemical shift of 3 ppm is that for CH_2_ protons of the anhydride cycle, which indicated the OSA reaction was complete. The area of the proton peak corresponding to the acyl group increased with the increase of the degree of substitution, and the intensity of the reducing chain ends of OSA–SGC decreased, which might be due to the fact that more reducing end of the chain need to be converted by OSA [23].

By controlling the ratio of OSA to SGC, OSA–SGC polymers with different degrees of substitution could be obtained. As can be seen from the Table 1, with the ratio of octenyl succinic anhydride/short glucan chain (OSA/SGC) increased from 25% to 50% and 100% respectively, the degree of substitution increased from 0.112 to 0.286 and 0.342 respectively. In the whole reaction process, the amount of OSA was the main influencing factor, but with the increase of the amount of OSA, the effect of the reaction temperature and reaction time on reaction efficiency was greater than that of the amount of OSA. With the increase of the amount of OSA, the efficiency of the reaction first increased and then decreased, there might be two reasons: Firstly, if there was no water in the system, the starch particles could not absorb water swelling, in the reaction process, OSA could not enter the interior of the particles, but too much water was not appropriate, too little was difficult to maintain the reaction, too much starch was easy to gelatinize, from this point of view, A certain volume of water should be maintained in the system. Secondly, OSA was a kind of lipophilic and hydrophilic substance, but too much and too little content would affect the contact between anhydride and the short glucan chain, which further affected the efficiency of the reaction.

### 3.2. Self-Assembly Behavior of Amphiphilic Starch Derivatives

In the current research, amphiphilic OSA–SGC polymers can be self-assembled into nanostructures in aqueous solution. In the process of self-assembly, CMC is an important factor to describe the self-aggregation behavior, indicating the formation and thermodynamic stability of nanoparticles. Since the amphiphilic polymer can only self-assemble to form micelle when the concentration of amphiphilic polymer is higher than the critical micelle concentration, CMC is an important parameter to characterize nano-micelle. The fluorescence intensity of pyrene increases with the increase of polymer concentration, and the fluorescence intensity changes most obviously at 373 nm and 383 nm, which fully reflects that the short glucan chain of octenyl succinate can form the hydrophobic structure, and the fluorescence probe pyrene moves to the hydrophobic region simultaneously, the emission spectrum will also change [27]. Therefore, the critical micelle concentration can be determined by the ratio I1/I3 of the highest energy band intensity of the first (373 nm) and the third (383 nm) in the pyrene emission spectrum.

In Figure 2, the critical micelle concentration CMC could be determined by the interception of the two straight lines in the longitudinal coordinates. When the concentration of the solution was low, the ratio of fluorescence intensity (I_1_/I_3_) basically did not change much, but with the increase of concentration and reaching CMC, the value of I_1_/I_3_ would decrease sharply, which also indicated the formation of nanostructure, and pyrene moved from the hydrophilic region to the hydrophobic region of micelle [28]. As shown in Table 2, when the degree of substitution of octenyl succinic acid on the short glucan chain increased from 0.112 to 0.342, the critical micelle concentration (CMC) decreased from 0.128 mg/mL to 0.0576 mg/mL. The results showed that the degree of substitution had a significant effect on the critical micelle concentration (CMC). When the degree of substitution increased slightly, CMC decreased obviously. The reason might be that the higher the degree of substitution of octenyl succinic acid short glucan chain (OSA–SGC), the stronger the hydrophobicity of micelle, and the movement of fluorescence probe pyrene would be more significant. The above results showed that after the formation of nano-micelle, the amphiphilic OSA–SGC polymer would form a hydrophobic core, so it could be used as a hydrophobic drug carrier [29].

### 3.3. Inclusion Complexes of Encapsulation Efficiency and Loading Content

As shown in Table 3, the encapsulation efficiencies (EE) of 62.06% (OSA_0.25_–SGC), 64.14% (OSA_0.5_–SGC) and 64.62% (OSA_1.0_–SGC) were high but not significantly different. Thus, higher DS of OSA–SGC–CUR did not affect the encapsulation efficiency so much. A relatively high EE could probably be due to the presence of amphipathic OSA, and the lipophilic properties of curcumin that made it easy for curcumin to be encapsulated. Chang [25] reported similar results. Aditya et al. [30] prepared an oil phase system consisting of stearic acid glyceride, oleic acid and lecithin as a wall material, with encapsulated curcumin. The obtained inclusion compounds had achieved an encapsulation efficiency of 78% ± 2%. The disordered lattice structure of glyceryl stearate appeared to the reason for increased EE. The loading content was found to be 6.21%, 6.43% and 6.57%, in the same order, indicating that these nanoparticles could be effectively used as a carrier of functional ingredients, which were otherwise insoluble in water.

### 3.4. DS on Curcumin Encapsulation, Particle Size, Zeta Potential and Polydispersity Coefficient

Although many researchers had analyzed the DS diversification to investigate the additives to OSA groups in starches after chemical modification, few teams had quantified the influence of OSA starches DS with curcumin encapsulation, particle size, zeta potential and polydispersity coefficient (PDI). The particle size affected the bioavailability of the guest molecules by affecting the retention time, the rate of dissolution and the behavior of the digestive enzymes in the gastrointestinal tract of the guest molecules, which played an important role in determining the bioavailability of the encapsulated molecules. Table 4 showed that the particle diameters were below 300 nm, while the nano-capsules had a particle size range of 1 to 1000 nm, indicating that the samples might be classified as nano-capsules. As the DS increased, the particle size of both OSA–SGC and OSA–SGC–CUR nano-spheres decreased significantly, mainly because the high-degree-of-substitution OSA–SGC had more hydrophobic groups that in turn forms tightly hydrophobic core during nano-micelles formation. The OSA–SGC loading with curcumin significantly changed the size distribution and zeta potential as DS increased and the relatively high DS (0.342) of OSA–SGC showed better curcumin encapsulation efficiency: These findings were in accord with some reported examples. For example, OSA starch encapsulated orange-peel oil had a better retention capability than the normal starch [31]. The polydispersity coefficient (PDI) of the nano-solutions remained below 0.2, indicating that the prepared OSA–SGC nano-spheres exhibited better dispersibility.

### 3.5. FTIR Analysis

The structural features of OSA–SGC, OSA–SGC–CUR and curcumin were established by the FTIR. The esterified SGC had a new structure with a stretching vibrational absorption at 1730 cm^−1^ (Figure 3C–E) and asymmetric stretching vibrations of the carboxyl groups (RCOO–) resulted in a new absorption at 1572 cm^−1^ [32,33]. These two peaks confirmed the successful introduction of OSA onto the SGC. The intensity of these two peaks increased with the degree of substitution accentuating addition of more OSA onto the SGC. These results were in agreement with Sun’s observations [25,34]. The characteristic absorption at 1730 cm^−1^ and 1572 cm^−1^ showed an increasing trend with the DS.

The hydroxyl group of curcumin had an absorption at 3503 cm^−1^ (Figure 4 and Figure 5), the C=O and C=C vibration at 1509 cm^−1^ and the stretching vibrations of benzene ring showed sharp absorption bands at 1605 cm^−1^. The characteristic peak at 1273 cm^−1^ was due to aromatic C–O stretching vibration, and peaks at 1026/856 cm^−1^ were attributed to C–O–C stretching vibrations of the curcumin [35]. The peak shape and intensity of the inclusion complex changed significantly compared to the physical mixture of OSA–SGC and curcumin. In the inclusion complexes of OSA–SGC–CUR, peaks were observed at 3392, 2931, 1730 and 1572 cm^−1^, matched well with those of OSA–SGC. In addition, only few characteristic peaks of curcumin were visible in the spectrum of the inclusion complex, which showed that C=O, C=C and the benzene ring of curcumin might be encapsulated into the core-shell structure of the OSA–SGC, but the curcumin physical mixtures exhibited both peaks of OSA–SGC and curcumin. These results were in agreement with those reported by Darandale [35]. The intensity of characteristic absorption peaks increased with DS [36]. These observations were consistent with the infrared results of the inclusion complex of rose essence embedded by β-cyclodextrin. The changes of peak shape, peak position and intensity could indicate whether the guest molecule entered the cavity and the interaction force [37].

### 3.6. X-ray Diffraction Analysis

The X-ray diffraction patterns of SGC, OSA–SGC and OSA–SGC–CUR were shown in Figure 5 and Table 5. The SGC displayed peaks at 14.4, 17.1, 19.4, 22.3 and 24.1°. Among them, the 14.4 and 19.4° arose from the V-type starch structure, and the 17.1, 22.3 and 24.1° from the B-type starch structure. Thus, SGC was composed of V- and B-starch crystallites and agreed with literature reports [22]. In Figure 6, the OSA substitution did not alter the crystalline diffractions pattern, especially for the DS 0.112 suggesting that the esterification might be occurrence in the amorphous region. However, at higher DS, e.g., 0.286 and 0.342, the diffraction patterns were deprived of 17.1, 22.3 and 24.1° peaks but with an increased 19.6° peak signifying the typical V-starch structure.

Curcumin possessed sharp and crystalline diffraction profiled at 7.9, 8.8, 12.1, 13.8, 15.8, 17.2 and 21.1°. The diffraction pattern from the physical mixture of curcumin and OSA–SGC (Figure 7) displayed these characteristics peaks but with weak intensity presumably due to shielding of curcumin by the OSA–SGC.

Interestingly, diffraction pattern from the complex was not simply an overlay of OSA–SGC and curcumin but was quite different (Figure 8). There were two definite peaks at 14.4 and 19.5° portraying a typical V-type starch structure [15]. The disappearance or weakening of characteristics peaks of OSA–SGC and curcumin demonstrated the complex formation.

### 3.7. TGA Analysis

Thermal stability of SGC, OSA–SGC and OSA–SGC–CUR was carried out through thermogravimetric analysis. Major loss of moisture was noticed before 100 °C (Figure 9). The second mass loss of the sample due to thermal decomposition took place between 190 and 340 °C. The OSA–SGC nanoparticles decomposed earlier than the SGC. As shown in Figure 10, a major mass loss of OSA_0.25_–SGC, OSA_0.5_–SGC, OSA_1.0_–SGC occurred at 225–316, 220–290 and 190–230 °C, respectively. The mass loss temperature was decreasing with the increase of DS. Generally, the higher the degradation temperature, the better the thermal stability. Thus, it appeared that the thermodynamic stability of the OSA–SGC was getting reduced with the OSA introduction. The addition of the hydrophobic group onto the SGC altered the hydrophobicity as well as the hydrogen bonding interactions between the SGCs and in-turn reduced the thermal stability. Similar occurrence was noticed with rice, wheat and potato starches [38]. The curcumin incorporation increased the overall thermal stability of OSA–SGC. The weight loss of OSA_0.25_–SGC–CUR, OSA_0.5_–SGC–CUR and OSA_1.0_–SGC–CUR occurred at 240–480, 238–470 and 235–450 °C, respectively. The trend to decrease the thermal stability with the increase of DS still continued but the subtle decrease could be mostly attributed to curcumin encapsulation. Comparing the temperature changes with OSA–SGC–CUR showed that the thermal stability of the inclusion compounds had improved compared to curcumin, and indeed agreed with Park’s [39] conclusion that encapsulated microspheres would improve the thermal stability.

### 3.8. Transmission Electron Microscopy (TEM)

To understand the morphology of the OSA–SGC nanostructures, spatial structure data had been collected using a high-resolution transmission electron microscope. Figure 11 depicted the nano-micelles formed by the OSA–SGC. The particle sizes of OSA_0.25_–SGC, OSA_0.5_–SGC and OSA_1.0_–SGC were about 15–20, 10–20 and 5–10 nm. The OSA_0.5_–SGCs formed small nanoparticles of size of 15–20 nm (Figure 11A). The reason might be that some small individual micelles reaggregated into these large polymers. In Figure 11B, most of the OSA_0.5_–SGCs formed ultra-small micelles, and in addition, could form large polymers [40]. Increasing the OSA to 0.5 further reduced the particles size of OSA_0.5_–SGC to 10–20 nm (Figure 11B). Very small particles of size 5–10 nm were observed, interestingly, with 1.0 of OSA (Figure 11C). The size of nano-micelle decreased with the increase of OSA content, which might be due to the increase of OSA content, which enhanced the hydrophobicity and made the core of the micelle more compact [41]. Thus, it appeared the DS had influenced the particle size but with subtle morphological changes.

### 3.9. Curcumin Release: In Vitro Simulation

The release from OSA–SGC nanoparticles in phosphate buffer at pH 6.8 and 7.4 was respectively shown in Figure 12. There were two distinct release stages: At the first stage, 0 to 10 h, the release was rapid signifying the burst phase from the residual curcumin adsorbed on the surface of the nanoparticles. Such a burst-relaxation-controlled release could be beneficial to gain suitable concentration of curcumin in the blood more rapidly and effectively. During the second stage, from 10–30 h, curcumin released slowly. After 30 h, the release rate decreased and reached a plateau but did not reach 100% [42]. This might be due to the influence of external environmental conditions, the change in the fluorophore function and adhesion between some of them might cause the curcumin content to decrease in a week-long experimental trial [43].

The release rate and curcumin quantity from the OSA–SGC–CUR nanoparticles might not be due to debranched short glucan chains but might depend on pH [44]. The release rate and extent in alkaline conditions were significantly higher than those in acidic conditions, indicating that the curcumin nanoparticle release rate in the simulated intestinal fluid environment was slower than that in the blood environment. Thus, the OSA–SGC–CUR nanoparticles could be fully hydrolyzed during the intestinal digestion, and thereby released curcumin in the small intestine. Curcumin was insoluble and was difficult to digest and to be absorbed in the human body [45]. It appeared that our OSA-–SGC–CUR nanoparticles could alleviate this difficulty by enhancing the curcumin dissolution and in-turn the bioavailability.

## 4. Conclusions

The OSA–SGC nanoparticles displayed high encapsulation efficiency of curcumin. However, DS had little effect on the total entrapment amount of curcumin. A high 64.62% of encapsulation efficiency was observed for the DS = 0.342. The lower release rate of the curcumin, from the OSA–SGC–CUR nanoparticles, in the simulated intestinal conditions, opened up novel opportunities to deliver health promoting and disease preventing compounds in the intestinal tract. These nanoparticles have the potential to deliver several water insoluble functional ingredients, and forms our future direction.

## Figures and Tables

**Figure 1 nanomaterials-09-01073-f001:**
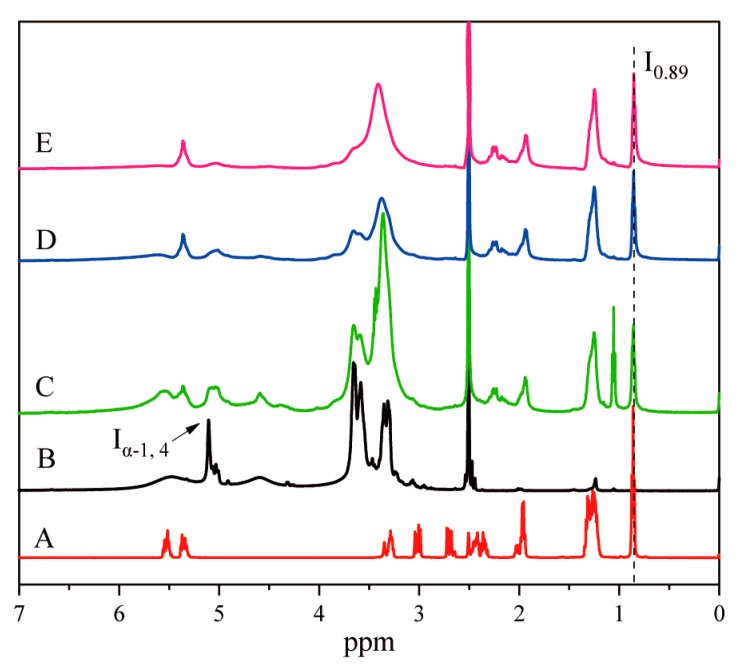
The hydrogen nuclear magnetic resonance (^1^H-NMR) spectra of OSA (A), short glucan chains (SGC; B), OSA_0.25_–SGC (C), OSA_0.5_–SGC (D) and OSA_1.0_–SGC (E).

**Figure 2 nanomaterials-09-01073-f002:**
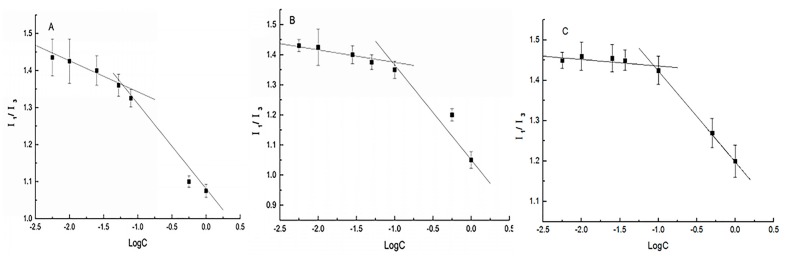
Critical micelle concentration (CMC) of OSA–SGC with a different degree of substitution (DS): (**A**) DS = 0.112, (**B**) DS = 0.286 and (**C**) DS = 0.342.

**Figure 3 nanomaterials-09-01073-f003:**
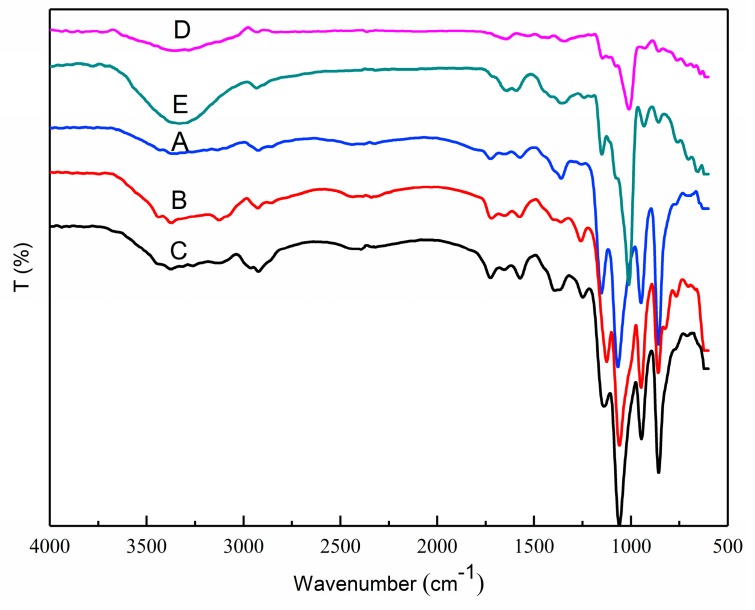
FTIR spectra: DS = 0.112 (A), DS = 0.286 (B), DS = 0.342 (C), native waxy maize starch (D) and SGC (E).

**Figure 4 nanomaterials-09-01073-f004:**
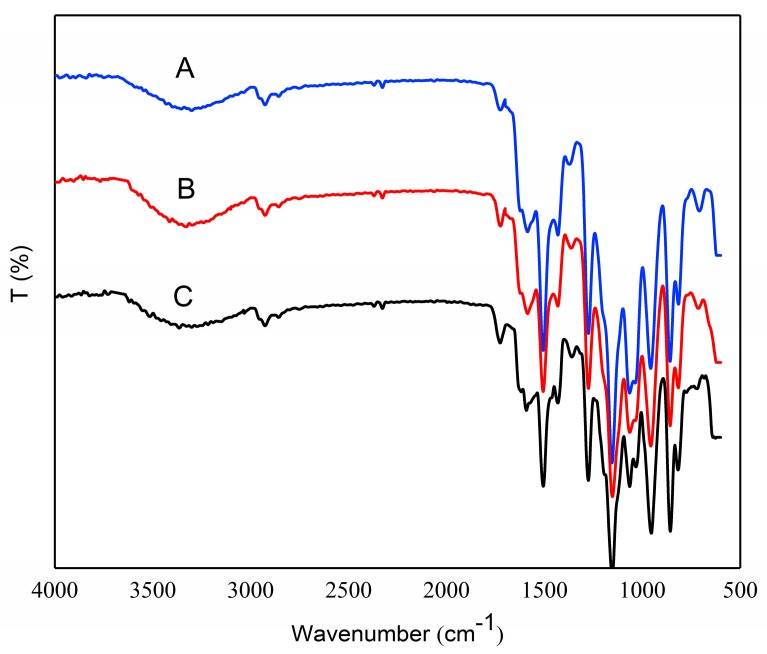
FTIR curves: OSA_0.25_–SGC–CUR—A, OSA_0.5_–SGC–CUR—B and OSA_1.0_–SGC–CUR—C mixture.

**Figure 5 nanomaterials-09-01073-f005:**
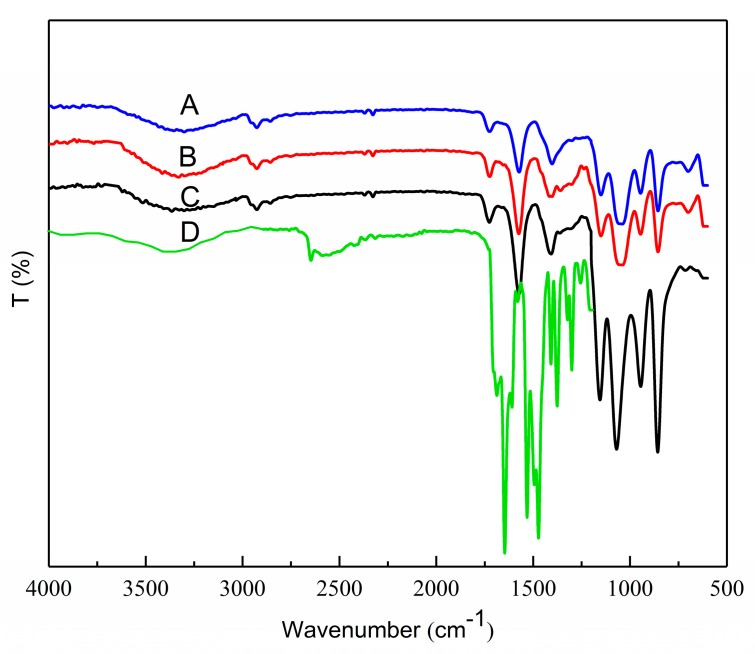
IR curves of inclusion complexes: OSA_0.25_–SGC–CUR—A, OSA_0.5_–SGC–CUR—B, OSA_1.0_–SGC–CUR—C and curcumin—D.

**Figure 6 nanomaterials-09-01073-f006:**
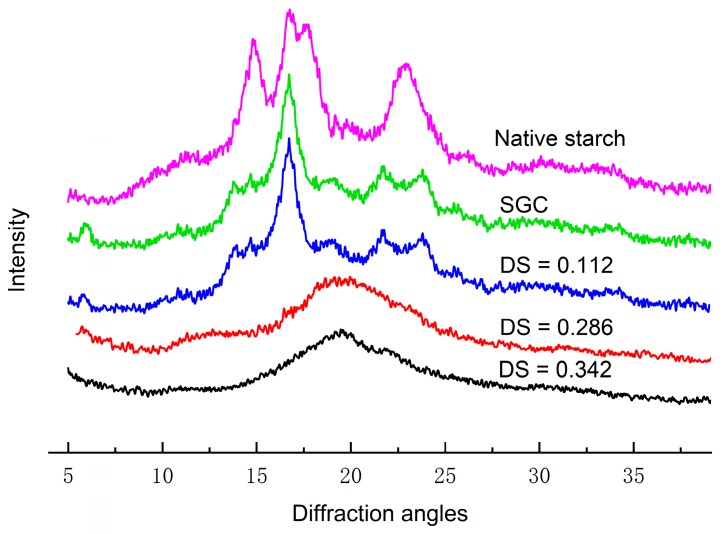
X-ray diffraction patterns of native waxy maize starch, SGC and OSA–SGC.

**Figure 7 nanomaterials-09-01073-f007:**
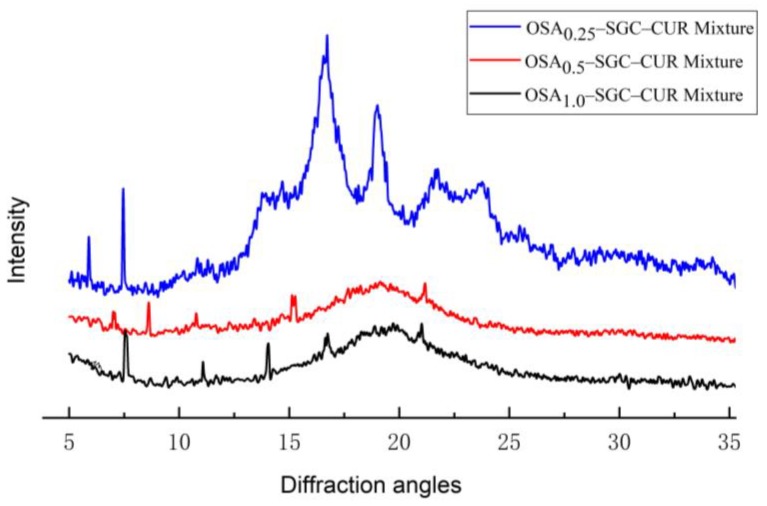
X-ray diffraction (XRD) patterns of the curcumin–short glucan chain physical mixture.

**Figure 8 nanomaterials-09-01073-f008:**
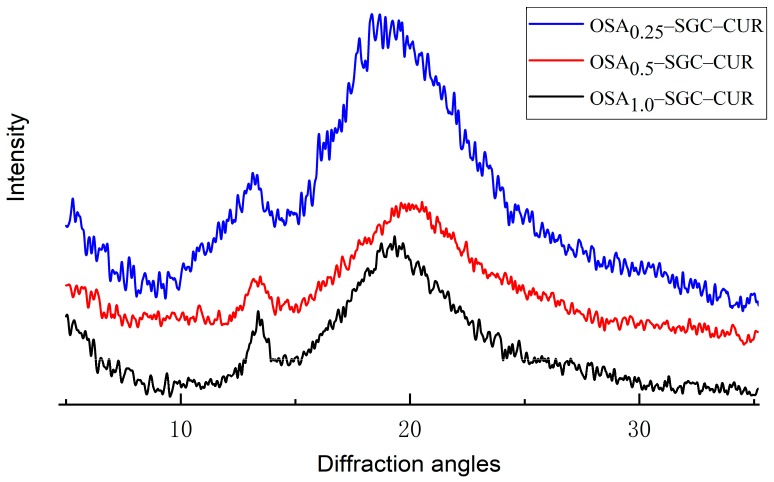
X-ray diffraction patterns of the curcumin and OSA–SGC–CUR inclusion complexes.

**Figure 9 nanomaterials-09-01073-f009:**
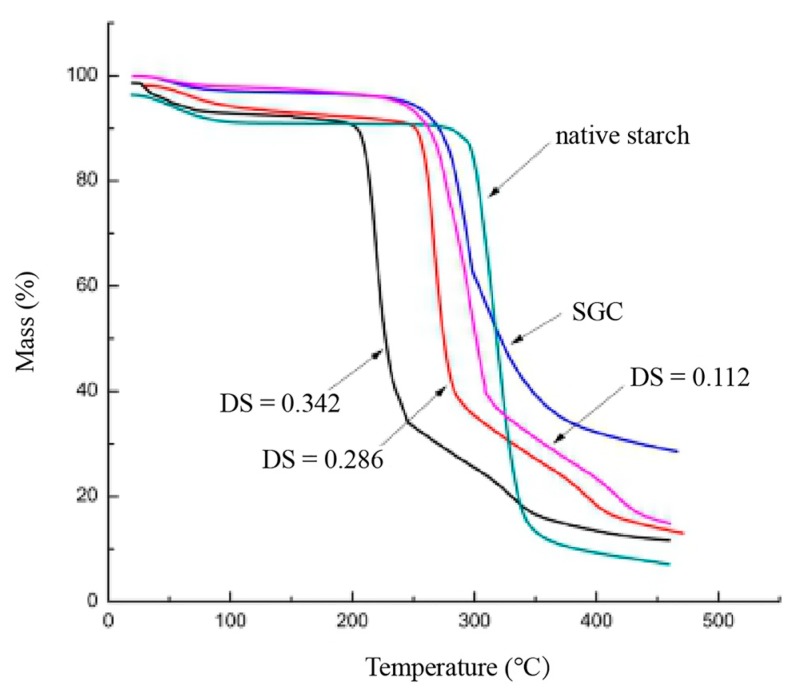
The TGA curves of SGC and OSA–SGC nanoparticles.

**Figure 10 nanomaterials-09-01073-f010:**
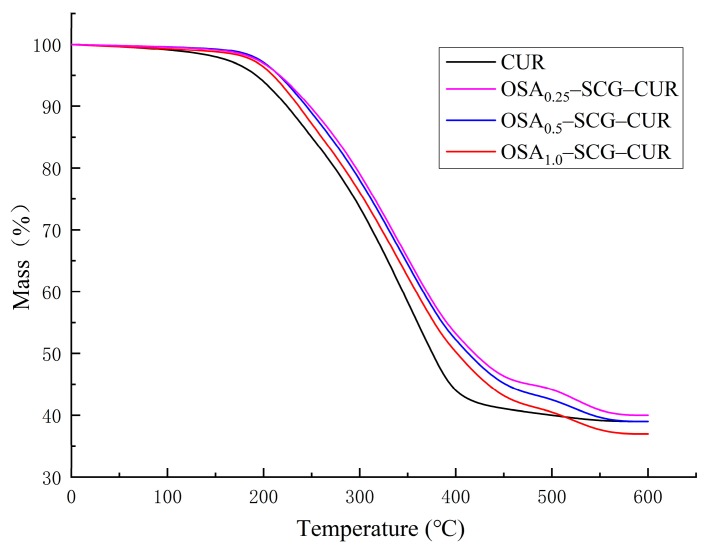
The TGA curves of curcumin and inclusion complexes.

**Figure 11 nanomaterials-09-01073-f011:**
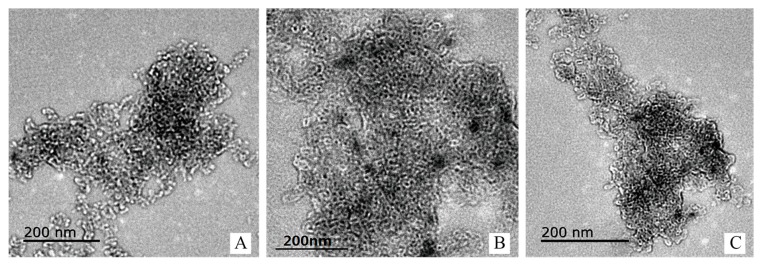
The TEM of SGC with different DS (OSA_0.25_–SG (**A**), OSA_0.5_–SGC (**B**) and OSA_1.0_–SGC (**C**)).

**Figure 12 nanomaterials-09-01073-f012:**
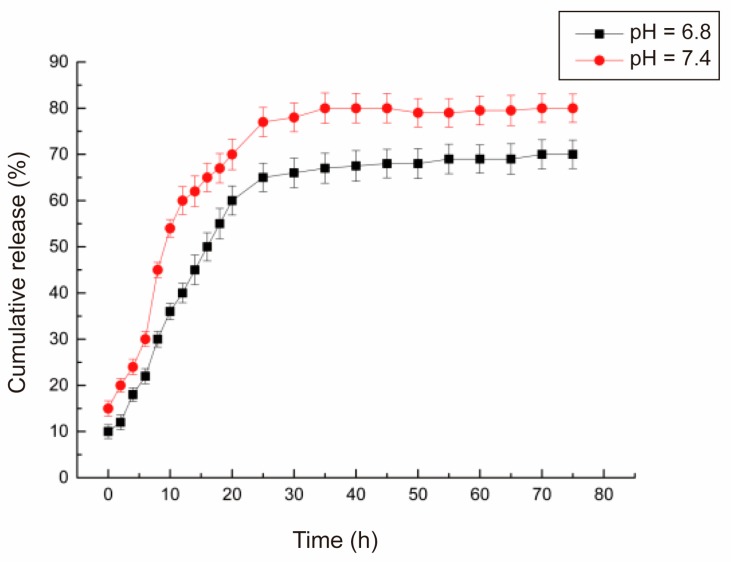
The release rate of curcumin in an intestinal (pH = 6.8) and blood (pH = 7.4) environments.

**Table 1 nanomaterials-09-01073-t001:** Effect of the ratio of OSA/SGC on degree of substitution (DS) and RE (reaction efficiency).

Sample	DS	RE
SGC	0	0
OSA_0.25_–SGC	0.112 + 0.010 ^a^	57.93% + 0.20 ^b^
OSA_0.5_–SGC	0.286 + 0.020 ^b^	73.97% + 0.28 ^c^
OSA_1.0_–SGC	0.342 + 0.020 ^c^	44.22% + 0.26 ^a^

Note: Different letters in the same column indicate significant differences, *p* < 0.05.

**Table 2 nanomaterials-09-01073-t002:** Critical micelle concentration of octenyl succinic anhydride short glucan chains with different degrees of substitution.

Sample	CMC
OSA_0.25_–SGC	0.128 mg/mL ± 0.0135 ^b^
OSA_0.5_–SGC	0.0746 mg/mL ± 0.0054 ^a^
OSA_1.0_–SGC	0.0576 mg/mL ± 0.0043 ^c^

Note: Different letters in the same column indicate significant differences, *p* < 0.05.

**Table 3 nanomaterials-09-01073-t003:** Encapsulation efficiency and loading content of curcumin in the OSA–SGC–CUR (curcumin) inclusion complexes.

Sample	Encapsulation Efficiency (%)	Loading Content (%)
OSA_0.25_–SGC–CUR	62.06 ± 1.31	6.21 ± 0.24
OSA_0.5_–SGC–CUR	64.14 ± 1.79	6.43 ± 0.13
OSA_1.0_–SGC–CUR	64.62 ± 1.52	6.57 ± 0.21

**Table 4 nanomaterials-09-01073-t004:** The particle size and the polydispersity coefficient (PDI) of OSA–SGC–CUR complexes.

Sample	Particle Size (nm)	PDI	Zeta (mV)
OSA_0.25_–SGC	184.67 ± 17.25 ^a^	0.152 ± 0.054 ^c^	−24.12 ± 1.23 ^bc^
OSA_0.5_–SGC	135.32 ± 10.35 ^a^	0.132 ± 0.041 ^a^	−26.84 ± 1.45 ^c^
OSA_1.0_–SGC	116.34 ± 8.32 ^bc^	0.127 ± 0.032 ^b^	−33.72 ± 2.01 ^a^
OSA_0.25_–SGC–CUR	287.64 ± 23.31 ^a^	0.210 ± 0.050 ^a^	−15.87 ± 1.01 ^a^
OSA_0.5_–SGC–CUR	236.46 ± 20.12 ^ab^	0.330 ± 0.030 ^ab^	−17.21 ± 1.32 ^b^
OSA_1.0_–SGC–CUR	192.35 ± 17.34 ^bc^	0.170 ± 0.030 ^ac^	−20.49 ± 1.45 ^c^

Note: Different letters in the same column indicate significant differences, *p* < 0.05.

**Table 5 nanomaterials-09-01073-t005:** The effect of OSA and curcumin encapsulation on the crystalline type of SGC.

Sample	Type
Waxy corn starch	A type
SGC	B + V type
OSA_0.25_–SGC	B + V type
OSA_0.5_–SGC	V type single spiral
OSA_1.0_–SGC	V type single spiral
OSA_0.25_–SGC–CUR	B + V type
OSA_0.5_–SGC–CUR	V type
OSA_1.0_–SGC–CUR	V type

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
