# Peer review of "Structural Characterization and Digestibility of Curcumin Loaded Octenyl Succinic Nanoparticles"

_nanomaterials, 2019, doi:10.3390/nano9081073_

Round 1
Reviewer 1 Report
The manuscript “Structural characterization and digestibility of curcumin loaded octenyl succinic nanoparticles” has been improved but some parts have to be better explained and written before the publication.
In the paragraph 2.3. Determination of degree of substitution there are some mistakes, for example:
“This experiment was carried out by hydrogen spectroscopy (Brook, Switzerland) at 600 MHZ, weighing 20 mg of raw starch and short glucan chains, respectively. And octenyl succinate short-glucan chain nanoparticles in a nuclear magnetic tube”
In addition it is not clear why the authors analyze raw starch. The nanoparticles have been prepared with short chain starch.
“ TFA-d6” doesn’t exist.
The sentence: what does the sentence “Ir-e represented the integral area of the proton at the end of the starch reduction” makes not sense. Can the author explain better?
Pag 3 in the equation describing the LC% is wrong. In the denominator is present the term “ quality of nanomicelles”. What does quality mean?
Pag.5 The experiment has not been properly designed in order to simulate intestinal release. Pancreatin and bile salts have to be added in the medium.
Pag. 5 line 197: “With the increase of the degree of substitution, the absorption peak at 5.11-5.25ppm became wider, which was due to the fact that the reaction of octenyl succinic anhydride was mainly carried out at O ≥ 2, which had been reported by Bai [26] et al.” it is not clear what is O ≥ 2.
In addition the ref 26 and 27 regard curcumin and not NMR determination of degree of substitution.
What do the authors mean with the sentence: “the strength of the end of the reduction chain decreased”?
Pag. 6 fig 1 is not clear, it is not possible to read the chemical shift scale.
Pag. 7: The authors write: “The encapsulation efficiencies(EE) of 62.06% (OSA0.25-SGC), 64.14% (OSA0.5-SGC), and 64.62%257 (OSA1.0-SGC)were high but not significantly different. Thus, higher DS of OSA-SGC-CUR did not 258 affect the encapsulation efficiency so much.”
And, in the Conclusions: “The OSA-SGC nanoparticles display high encapsulation efficiency of curcumin. The DS influences the total curcumin encapsulation amount.”
The two sentences are in contradiction. In general the conclusions should be extended.
Reviewer 2 Report
The quality of the manuscript was significantly improved after revision. Please add description of a,b,c, symbols in the captions of Tables 1-4.
Reviewer 3 Report
Compared with the originally submitted manuscript, this version looks better, although a number of changes should be made before it is acceptable for publication.
In particular, most of the newly added parts still have poor grammar, and my concerns regarding the significant digits have not been addressed. For example, in table 4: particle size 184.67nm with a reported error of 17.25. How could the authors measure the ".67" with an error of 17? and how could they measure the "17.25" error? I repeat the standard convention REQUIRES that measurements are reported aprropriately, e.g. "180 +- 20" or "185 +-17" at worst.
The characterization seems better, but still not completely appropriate. In Figure 1 there are several peaks which are not understandable, especially in spectra c and d. I would like to be more precise, but the quality of the figure is so low that it is not possible even to read the scale!
Additionally, a apectrum of OSA should be provided in the same figure, to ease comparison.
Round 2
Reviewer 3 Report
even though the error digits are still not completely consistent, I believe the manuscript could be acceptable for publication
This manuscript is a resubmission of an earlier submission. The following is a list of the peer review reports and author responses from that submission.
Round 1
Reviewer 1 Report
The manuscript “Structural characterization and digestibility of curcumin loaded octenyl succinic nanoparticles” describes the preparation and characterization of nanoparticles made of short glucan chains (SGC) esterified with a succinic acid moiety. The manuscript is poorly written and often not clear. In particular, the introduction has to be rewritten with the help of an English native or expert in this language. In addition:
Pag1 line 36-38. The sentence: “In this regard, addition of groups such as octenyl succinic anhydride (OSA) are being explored , OSA displays excellent emulsification properties, and good encapsulation efficiency toward many kinds of sensitive and insoluble functional molecules” is not only poorly written but also imprecise from a scientific point of view. In fact the SGC react with OSA to give succinic ester derivatives; the OSA is not present in the final polymer and cannot act as emulsifier.
Pag 2 line 73. “waxy corn starch (5%, 15%, 25%) slurry (100ml Phosphate buffer solution/pH=5.0)” What is the unit of concentration used?
Pag 2 line 78. “OSA (100% based on starch)”: what do the authors mean?
Pag.2 line 82. The authors have to describe how the DS has been determined.
Pag 3 line 96. “Curcumin was dissolved in 400 mL methanol at a concentration of 1000 g/mL under constant stirring. “There is mistake in the concentration value.
Pag 3 line 96-107. This part is confusing. The authors prepared OSA-SGC-CUR and claim to have measured the encapsulation efficiency but the method used is not clear. Can they describe accurately all the steps?
Pag 3 The equation describing the LC% is wrong. In the denominator is present the term “ quality of nanomicelles”
Pag. 4 line 142-148. The experiment has not been properly designed in order to simulate intestinal release. Pancreatin and bile salts have to be added in the medium. In addition, if the authors want to study the stability of nanoparticles in GI tract, they have to add a part regarding the stability in a system simulating stomach environment.
Pag.5 line 167-85. The paragraph is poorly written and sometimes inaccurate. The statement: ”OSA-SGC has more hydrophobic groups” do not take into account that the reaction with OSA inserts in the polymer COOH groups.
Pag. 6 In Fig.2 the spectrum of OSA-SGC is missing
Reviewer 2 Report
Curcumin is an oil soluble pigment, practically insoluble in water at acidic and neutral pH, soluble in alkali. It is unstable in alkaline conditions and in the presence of light. Because of the pharmacological efficacy and safety of curcumin, it has been investigated extensively in a wide range of research areas. However, its oral bioavailability is low due to poor solubility in water and the instability under physiological and alkaline pH conditions. This is why various formulations have been developed to improve its aqueous solubility and oral bioavailability. The presented paper continues the search of new curcumin carriers. However, the quality of the manuscript does not allow me to recommend it for publication in the present form. The following questions should be addressed before next submition.
1) English should be considerably improved. Here are just some examples from the Abstract:
“Curcumin is intrinsic anti-cancer, anti-inflammatory, anti-obesity and insolubility, Herein,..”; “the an amphiphilic biopolymer”; “one possible ways”, and so on.
2) Lines 55-60, add more refs on curcumin encapsulation with different carriers, for example (Zhang at al. Preparation, physicochemical and pharmacological study of curcumin solid dispersion with an arabinogalactan complexation agent. International Journal of Biological Macromolecules. doi:10.1016/j.ijbiomac.2019.01.079; Zhang et al, Preparation of curcumin self-micelle solid dispersion with enhanced bioavailability and cytotoxic activity by mechanochemistry, Drug Delivery 25(1) (2018) 198-209) and refs therein.
3) Lines 66-71 – Remove!
4) Line 103 “Curcumin was dissolved in 400 mL methanol at a concentration of 1000 g/mL” – correct concentration.
5) Eq. 2: “The quality of nano micelles” – may be quantity?
Why micelles? Where is evidence that yours nanoparticles are micelles?
6) Lines 181-185, “Table 2 shows that the particle diameters were all below 300 nm, while the nano-capsules had a particle size range of 1 to 1000 nm, indicating that the samples may be classified as nano-capsules”. Please define clearly what particles you have, nanoparticles, nano-capsules or nano-micelles! With strong evidence.
7) Line 191 and 272-283, “OSA-SGC-CURs have a uniform particle size”, this conclusion is wrong, according your TEM experiment, you see some not uniform aggregates with size 200-1000 nm.
8) Lines 191-202, figure 1 does not allow to estimate solubility enhancement. Use UV spectroscopy or HPLC to calculate solubility, and then compare with free curcumin solubility (1.73 mg/l [Zhang at al. Preparation, physicochemical and pharmacological study of curcumin solid dispersion with an arabinogalactan complexation agent. International Journal of Biological Macromolecules. doi:10.1016/j.ijbiomac.2019.01.079; Zhang et al, Preparation of curcumin self-micelle solid dispersion with enhanced bioavailability and cytotoxic activity by mechanochemistry, Drug Delivery 25(1) (2018) 198-209]).
Reviewer 3 Report
The manuscript by Feng et al. describes the formulation and characterization of curcumin-loaded nanoparticles. The manuscript is dealing with an extensively investigated, yet still interesting, topic (curcumin encapsulation), but it should be improved before being publishable.
In particular:
1) methods lack some detail. It is not present (or at least not evident) which strategy was used to determine the actual OSA derivatization grade during reaction with SGC. Given that some side reactions could be present, the amount of unreacted (or hydrolyzed) material should be evaluated and a more appropriate derivatization degree should be provided.
2) The english is difficult to read and full of typos and mis-spelled sentences (e.g. "result is the same with SUN" row 202, "this result is same with." row 291, and so on. This makes difficult even to understand the meaning of the article. The authors should bring the english level at least to an acceptable level by having the manuscript checked by a native english speaker.
3) errors are indicated with a wrong number of digits in the manuscript. The authors should correct them according to international conventions (i.e. significant digits in the measurement should stop at the first digit of the error)
4) I do not think that figure 1 is necessary. It is merely qualitative, not further characterized, and it is not clear whether the amount of curcumin in the first flask is the same as in the others (I think it isn't).